# Chemical Compositions of Walnut (*Juglans* Spp.) Oil: Combined Effects of Genetic and Climatic Factors

Hanbo Yang [1], Xu Xiao [1,2], Jingjing Li [1], Fang Wang [1], Jiaxuan Mi [1], Yujie Shi [1], Fang He [1], Lianghua Chen [1], Fan Zhang [3] and Xueqin Wan [1,*]

[1] Forestry Ecological Engineering in the Upper Reaches of the Yangtze River Key Laboratory of Sichuan Province & National Forestry and Grassland Administration Key Laboratory of Forest Resources Conservation and Ecological Safety on the Upper Reaches of the Yangtze River & Rainy Area of West China Plantation Ecosystem Permanent Scientific Research Base, Institute of Ecology & Forestry, Sichuan Agricultural University, Chengdu 611130, China; yanghanbo6@sicau.edu.cn (H.Y.); 2020304074@stu.sicau.edu.cn (X.X.); 2020304092@stu.sicau.edu.cn (J.L.); 2020204014@stu.sicau.edu.cn (F.W.); 2020104002@stu.sicau.edu.cn (J.M.); 2019104006@stu.sicau.edu.cn (Y.S.); 14686@sicau.edu.cn (F.H.); chenlh@sicau.edu.cn (L.C.)
[2] Sichuan Surveying and Planning Institute of Forestry and Grassland, Chengdu 311400, China
[3] College of Landscape Architecture, Sichuan Agricultural University, Chengdu 611130, China; 13305@sicau.edu.cn
[*] Correspondence: wanxueqin@sicau.edu.cn; Tel.: +86-18281366168

**Abstract:** Walnut oil is a high-value oil product. Investigation of the variation and the main climatic factors affecting the oil's chemical composition is vital for breeding and oil quality improvement. Therefore, the fatty acid, micronutrients, and secondary metabolites compositions and contents in walnut oil were determined in three species: *Juglans regia* L. (common walnut), *J. sigillata* Dode (iron walnut), and their hybrids (*Juglans sigillata* Dode × *J. regia* L.), which were cultivated aat different sites. The major fatty acids were linoleic (51.39–63.12%), oleic (18.40–33.56%), and linolenic acid (6.52–11.69%). High variation in the contents of fatty acids, micronutrients, and secondary metabolites was found between both species and sites. Interestingly, myristic, margaric, and margaroleic acid were only detected in the hybrids' walnut oil, yet α-tocopherol was only detected in common and iron walnut oil. Climatic factors significantly affected the composition and content of fatty acid, whereas δ-tocopherol was mostly dependent on the genetic factors. The average relative humidity explained the most variation in the fatty acids, micronutrients, and secondary metabolites, which showed a significant positive and negative effect on the monounsaturated fatty acids and polyunsaturated fatty acids, respectively. These findings contribute to the provision of better guidance in matching sites with walnut trees, and improvement of the nutritional value of walnut oil.

**Keywords:** walnut oil; fatty acid; micronutrients; secondary metabolites; variation; genetic effect; climatic factors

## 1. Introduction

Walnut (*Juglans* spp.) is one of the four famous tree nuts (walnut, almond, chestnut, and cashew) that have been consumed as rich nutritious food in many countries around the world [1]. Walnut oil is a high-value oil product that is used widely in food and health care [2]. Walnut oil is naturally rich in polyunsaturated fatty acids (PUFAs), mainly linoleic and linolenic acids, and is consequently poor in monounsaturated fatty acids (MUFAs), represented by oleic acid, and saturated fatty acids (SFAs) [3–5]. The fatty acid profile indicates that nutritionally, unsaturated fatty acid has a significant effect on the regulation of blood lipid, cleaning of thrombus, and immunoregulation [6]. Walnut oil is also rich in micronutrients and secondary metabolites, such as tocopherol, flavone, and polyphenols, which have been reported to exhibit numerous beneficial effects, such as antidiabetic, antioxidative, anti-inflammatory, and anti-proliferative effects in cancer [7,8]. With the

increasing emphasis on health care, walnut oil has become more popular among consumers for its particular health functions [2]. The nutritional value of walnut oil is mainly related to the species and genotypes of *Juglans*, and the environment factors [2,6]. Many researchers that have described the major and minor compositions of vegetable oils, which have been produced and published in standard texts [1–3,6,9–12]. The oil composition and content are determined by genetic control of plants [13]. For instance, the concentrations and compositions of tocopherol in almond oil are under genetic control [10]. Significant variation in the linoleic, linolenic, and oleic acid contents also exists naturally due to the genotypes in walnut oil [14]. Despite genetic control, environmental factors, such as latitude, temperature, and drought, may also affect the oil composition and concentration [3]. Much of the variation recorded in the fatty acid profile was associated with the cultivated site [14]. An increase in the oleic/linoleic acid rate with increasing temperature has been widely reported in several oil crops, such as sunflower [15], soybean [16], and walnut [3]. Hot summers also induce higher concentrations of tocopherol in almond and many other plant species [17]. However, little information has been reported about the combined effects of environmental factors during the maturing stage on different species or genotypes of walnut oil, such as temperature, rainfall, altitude, and latitude, etc.

Changes in the oil composition and content are currently the goal of many oilseed crop breeding programs [3]. The genus *Juglans* includes approximately 21 species that are widely distributed around the world [2,18]. Common walnut (*J. regia* L.) and iron walnut (*J. sigillata* Dode) are the two main species that are cultivated for nut production in China [19]. The two species can be used to extract and produce walnut oil. There are significant differences in the lipid composition and minor composition contents between common walnut and iron walnut [2]. Mating between the common walnut and iron walnut is compatible, and their hybrids are also widely cultivated in China. The hybrids may have a wider range of cultivation and higher quality of oil than the parents according to heterosis. There are different interaction strength effects of the genetic or environmental factors on the compositions and concentrations of vegetable oil. For instance, the site was the main effect on γ-tocopherol and δ-tocopherol, but for α-tocopherol, the site effect was dependent on the genotype [10]. The walnut species may affect the fatty acid composition, and the temperature affects the fatty acid content in walnut oil [2,3]. Therefore, comprehensive analysis of the differences in the fatty acid composition and content of walnut oil between species/genotypes and cultivation sites is important for breeding programs and cultivation (matching sites with species or varieties). Consequently, the aim of this work was (i) to distinguish walnut oil from different species/genotypes, (ii) determine the effect of genotypes and climatic factors on walnut oil, and (iii) evaluate the effect of the main climatic factors during nut development on the fatty acid.

## 2. Materials and Methods

### 2.1. Plant Materials and Experimental Area

The walnut samples consisted of *Juglans regia* L. (Common walnut), *J. sigillata* Dode (Iron walnut), and *J. sigillata* Dode × *J. regia* L. at nine cultivation sites. The commercial cultivar was 'Yanyuanzao' (a hybrid of *J. sigillata* Dode × *Juglans regia* L., widely cultivated in China), which was cultivated at six sites (the main distribution area of 'Yanyuanzao') under Hengduan Mountains. *J. regia* was cultivated at three sites, and *J. sigillata* Dode was cultivated at two sites. Information of the accessions and cultivation sites is shown in Table 1. In 2019, 3 replicates of 50 nuts were randomly collected from around the canopy of a healthy tree after open pollination for each accession at each site. The meteorological data (minimum, average, and maximum daily temperature; accumulated precipitation; sunshine duration; average and minimum relative humidity during fruit development) for each site were obtained from meteorological stations at the cultivated site.

**Table 1.** Geographic site and cultivated species/genotypes of the walnut samples.

| Cultivation Site | Species | Longitude/° | Latitude/° | Altitude/m |
|---|---|---|---|---|
| Batang, Ganzi, China (RBT) | Common walnut (*J. regia* L.) | 99.013 | 29.778 | 2493 |
| Derong, Ganzi, China (RDR) | Common walnut (*J. regia* L.) | 99.376 | 29.047 | 3196 |
| Jiulong, Ganzi, China (SJL) | Iron walnut (*J. sigillata* Dode) | 101.723 | 28.528 | 2304 |
| Jiulong, Ganzi, China (SRJL) | Hybrids (*J. sigillata* Dode × *J. regia* L.) | | | |
| Leibo, Liangshan, China (SLB) | Iron walnut (*J. sigillata* Dode) | 103.434 | 28.258 | 989 |
| Leibo, Liangshan, China (SRLB) | Hybrids (*J. sigillata* Dode × *J. regia* L.) | | | |
| Xiangcheng, Ganzi, China (RXC) | Common walnut (*J. regia* L.) | 99.467 | 29.087 | 2814 |
| Dechang, Liangshan, China (SRDC) | Hybrids (*J. sigillata* Dode × *J. regia* L.) | 102.017 | 27.050 | 1596 |
| Luding, Ganzi, China (SRLD) | Hybrids (*J. sigillata* Dode × *J. regia* L.) | 102.017 | 29.050 | 1412 |
| Mianning, Liangshan, China (SRMN) | Hybrids (*J. sigillata* Dode × *J. regia* L.) | 102.683 | 28.033 | 2004 |
| Yanyuan, Liangshan, China (SRYY) | Hybrids (*J. sigillata* Dode × *J. regia* L.) | 101.050 | 27.033 | 2527 |

Note: Common walnut and iron walnut were cultivated by seedlings, *J. sigillata* Dode × *J. regia* L. was cultivated by grafting with a commercial cultivar of 'Yanyuanzao'.

*2.2. Fatty Acid Composition*

The nuts were immediately transported to the laboratory after harvest, dried in an oven at 45 °C for 3 days, and dehusked using a hammer. The walnut oil was extracted from the kernels using the Soxhlet method [20]. The fatty acid was analyzed using an 8890 gas chromatograph (GC) (Agilent, Shanghai, China) and a CP-Sil 88 FAME capillary column (0.20 μm, 100 m × 0.25 mm, Agilent, Santa Clara, CA, USA). The operating conditions were as follows: nitrogen as the carrier gas with a linear velocity of 0.7 mL/min, flame ionization detector (FID) temperature of 280 °C and inlet temperature of 270 °C, split ratio of 100:1, and injection volume of 1.0 μL. The oven was held at 100 °C for 13 min and then programmed at 10 °C/min to 180 °C and held for 6 min, then programmed at 1 °C/min to 200 °C and held for 20 min, and finally increased to 230 °C at 4 °C/min and held for 10.5 min. The samples were identified by comparing the retention times of the sample peaks with those of a mixture of FAME standards. The fatty acid contents were expressed as the relative in terms of the percentage of individual fatty acids.

*2.3. Micronutrients and Secondary Metabolites Determination*

In total, 4.00 g of oil was weighted and diluted with 4 mL of *n*-hexane in a 10 mL volumetric flask and the content of tocopherols determined by high-performance liquid chromatography [21]. The ultraviolet detector was performed using high-performance liquid chromatography (LC-20A, Shimadzu, Tokyo, Japan) installed with a Waters Spherisorb Silica Column (250 mm × 4.6 mm, 5 μm). The injection volume of the sample was 10 μL, the column temperature was 20 °C, and the mobile phase was methanol with a rate of 0.8 mL/min. A determining wavelength of 294 nm was used. Through comparison of the standards, α-, β-, γ-, and δ-tocopherols were identified and quantified. The tocopherol compositions were the mean values of three replicates from each sample and were expressed as mg/kg oil. The polyphenols and flavone contents of walnut oil were determined using Folin's reagent and the aluminum nitrate-sodium nitrite colorimetric method, respectively [20]. The absorbance of the solution was measured at 510 nm for flavone and 765 nm for polyphenols. The flavone and polyphenols contents were the mean values of three replicates from each sample and were expressed as mg/kg oil.

*2.4. Statistical Analysis*

Statistical analysis was performed using R version 4.0.5 (https://www.r-project.org/, accessed on 31 December 2021). The functions of the Shapiro test and Bartlett test were used to calculate the normal distribution and the homogeneity of variance test. Then, one-way ANOVA was used to determine the difference in the chemical composition of walnut oil between different cultivation sites. The mean separation was assessed with the LSD test at $p \leq 0.05$. Further, two-way ANOVA based on the data of two sites (Jiulong and Leibo) of two species (iron walnut and hybrids) was used to explore the effect of species, cultivation

sites, and the species–sites interaction on the chemical composition and content of walnut oil (Table 1). Pearson correlation coefficients between the chemical characteristics were calculated ($\alpha = 0.05$). The correlation between sites/species was assessed by Pearson's test according to the chemical compositions of walnut oil. The correlation between the variation in the chemical composition of walnut oil and climatic factor variables was calculated by linear regression analysis. Further, to assess the relative importance of each climatic factor to explain the variation in walnut oil, full subset regression analysis and multiple regression models were implemented using ordinary least squares (OLS). All variables of the climatic factors were standardized before conducting the regression analysis.

## 3. Results

### 3.1. Fatty Acid Composition

Data on oil the fatty acid compositions and contents are reported in Figure 1. In all samples, the oil was mainly composed of five fatty acids: palmitic acid (C16:0, 4.01–5.41%), stearic acid (C18:0, 2.41–3.10%), oleic acid (C18:1, 18.40–33.56%), linoleic acid (C18:2, 51.39–63.12%), and linolenic acid (C18:3, 6.52–11.69%). Polyunsaturated acids (PUFAs) were the main group of fatty acids in the walnut oil in all three species, followed by monounsaturated fatty acids (MUFAs). The results of the one-way ANOVA showed that the content of PUFAs, MUFAs, linoleic, and linolenic acid were significantly different in the three walnut species. The contents of linoleic and linolenic acid in the hybrids (*J. sigillata* × *J. regia*) were significantly higher than common walnut (*J. regia*). Therefore, the PUFAs content of the hybrids was significantly higher than common walnut ($F = 27.62$, $p < 0.001$), which would be due to the heterosis of iron and common walnut. Oleic acid (C18:1) was the major MUFA, and the content of oleic acid in common walnut was significantly higher than that in hybrids and iron walnut ($F = 9.81$, $p < 0.001$). Notably, there was a negative correlation between the content of oleic (C18:1) and linoleic acid (C18:2) ($r = -0.979$, $p < 0.001$) (Figure 2). PUFA showed a significant negative correlation with the content of oleic acid (C18:1) and MUFAs but a significant positive correlation with the content of linoleic acid (C18:2). A significant positive correlation between the content of margaroleic acid (C17:1) and myristic (C14:0) and margaric acid (C17:0) was observed. There was a significant negative correlation between the content of eicosenoic acid (C20:1) and palmitic (C16:0) and palmitoleic acid (C16:1). Saturated fatty acids (SFAs) were the lesser group in walnut oil, ranging from 6.91% to 8.91%, with palmitic acid (C16:0) and stearic acid (C18:0) the two main SFAs present, totaling, on average, 4.31% and 2.63%, respectively (Figure 1). It should be noted that the hybrids' walnut oil contained some minor fatty acids, such as myristic acid (C14:0, 0.01–0.02%), margaric acid (C17:0, 0.02–0.03%), and margaroleic acid (C17:1, 0.02%), that were not detected in the common and iron walnut oil. We studied the effects of species, climatic factors, and the genotype–environment interaction on the fatty acid composition and content based on the design of two cultivation sites (Jiulong and Leibo) of two species (iron walnut and hybrids). The results of two-way ANOVA showed significant differences in the linolenic acid (C18:3) content between the sites and species (Table 2). A significant difference between species was shown regarding margaroleic acid (C17:1) ($p < 0.001$). The content of myristic acid (C14:0) was significantly influenced by species, cultivation sites, and the species–sites interaction.

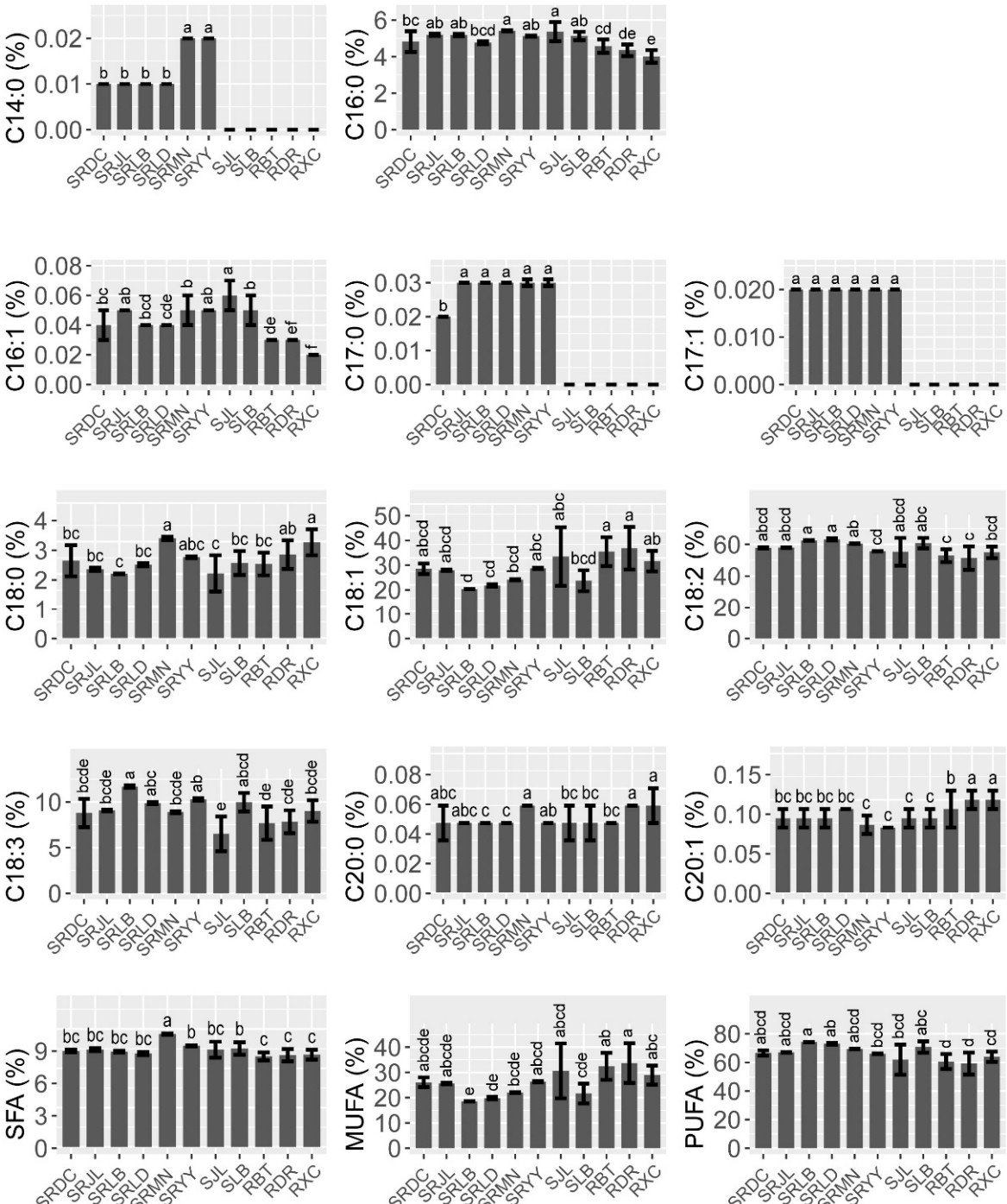

**Figure 1.** Fatty acid composition and contents of walnut oils. Note: Values are the mean ± standard deviation, and different letters indicate significant differences ($p < 0.05$) between different species/sites. SFAs, saturated fatty acids (C14:0 + C16:0 + C17:0 + C18:0 + C20:0); MUFAs, monounsaturated fatty acids (C:16:1 + C17:1 + C18:1 + C20:1); PUFAs, polyunsaturated fatty acid (C18:2 + C18:3), the same as below.

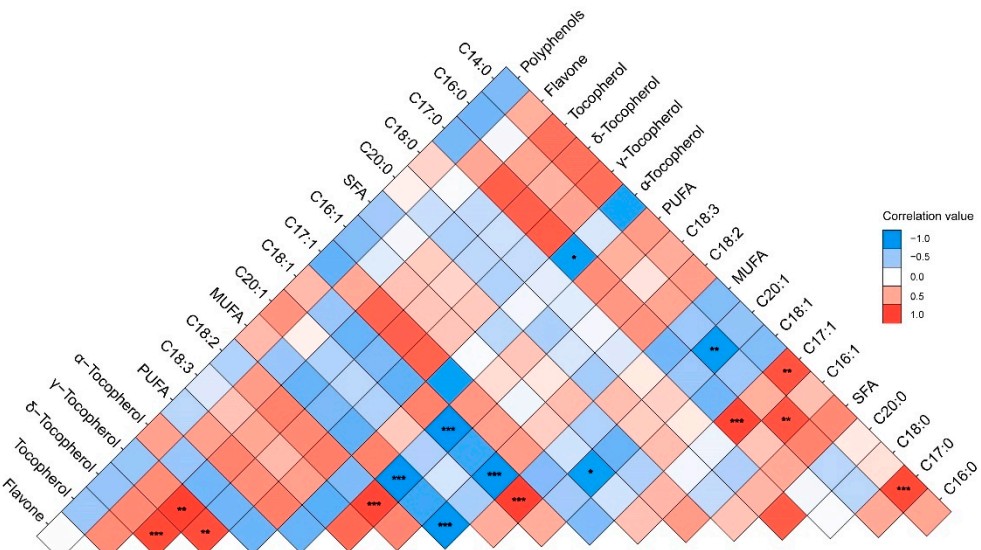

**Figure 2.** The correlation between the chemical characteristics of walnut oil. Note: *, **, and ***
indicate significant correlations (*p* < 0.05, 0.01, and 0.001) between the 19 chemical characteristics of
walnut oil, respectively.

**Table 2.** Two-way analysis of variance (two-way ANOVA) of fatty acid.

| Source | Sum of Square | Mean Square | *F* Value | Source | Sum of Square | Mean Square | *F* Value |
|---|---|---|---|---|---|---|---|
| C14:0 | | | | C18:1 | | | |
| Species | $4.51 \times 10^{-4}$ | $4.51 \times 10^{-4}$ | 3464.16 *** | Species | 19.50 | 19.50 | 0.474 |
| Sites | $1.40 \times 10^{-6}$ | $1.40 \times 10^{-6}$ | 10.74 * | Sites | 179.00 | 179.00 | 4.348 |
| Species × Sites | $1.40 \times 10^{-6}$ | $1.40 \times 10^{-6}$ | 10.74 * | Species × Sites | 1.50 | 1.50 | 0.037 |
| Residuals | $1.00 \times 10^{-6}$ | $1.00 \times 10^{-7}$ | | Residuals | 330.00 | 41.20 | |
| C16:0 | | | | C20:1 | | | |
| Species | $1.55 \times 10^{-1}$ | $1.55 \times 10^{-1}$ | 1.551 | Species | $1.96 \times 10^{-4}$ | $1.96 \times 10^{-4}$ | 1.69 |
| Sites | $2.64 \times 10^{-1}$ | $2.64 \times 10^{-1}$ | 2.654 | Sites | $3.10 \times 10^{-6}$ | $3.10 \times 10^{-6}$ | 0.027 |
| Species × Sites | $2.44 \times 10^{-1}$ | $2.44 \times 10^{-1}$ | 2.449 | Species × Sites | $2.59 \times 10^{-5}$ | $2.59 \times 10^{-5}$ | 0.223 |
| Residuals | $7.97 \times 10^{-1}$ | $9.96 \times 10^{-2}$ | | Residuals | $9.28 \times 10^{-4}$ | $1.16 \times 10^{-4}$ | |
| C18:0 | | | | MUFA | | | |
| Species | $5.20 \times 10^{-3}$ | $5.20 \times 10^{-6}$ | 0.073 | Species | 19.30 | 19.30 | 0.467 |
| Sites | $3.85 \times 10^{-2}$ | $3.85 \times 10^{-2}$ | 0.544 | Sites | 180.00 | 180.00 | 4.36 |
| Species × Sites | $1.82 \times 10^{-1}$ | $1.82 \times 10^{-1}$ | 2.569 | Species × Sites | 1.50 | 1.50 | 0.037 |
| Residuals | $5.66 \times 10^{-1}$ | $7.08 \times 10^{-2}$ | | Residuals | 330.00 | 41.30 | |
| C20:0 | | | | C18:2 | | | |
| Species | $9.18 \times 10^{-6}$ | $9.18 \times 10^{-6}$ | 0.317 | Species | 1.41 | 1.41 | 0.052 |
| Sites | $7.72 \times 10^{-5}$ | $7.72 \times 10^{-5}$ | 2.662 | Sites | 73.70 | 73.70 | 2.694 |
| Species × Sites | $2.64 \times 10^{-6}$ | $2.64 \times 10^{-6}$ | 0.091 | Species × Sites | $4.90 \times 10^{-1}$ | $4.90 \times 10^{-1}$ | 0.018 |
| Residuals | $2.32 \times 10^{-4}$ | $2.90 \times 10^{-5}$ | | Residuals | 219.00 | 27.30 | |
| SFA | | | | C18:3 | | | |
| Species | $6.32 \times 10^{-2}$ | $6.32 \times 10^{-2}$ | 2.425 | Species | 12.60 | 12.60 | 8.001 * |
| Sites | $1.07 \times 10^{-1}$ | $1.07 \times 10^{-1}$ | 4.12 | Sites | 26.60 | 26.60 | 16.921 ** |
| Species × Sites | $4.20 \times 10^{-3}$ | $4.20 \times 10^{-3}$ | 0.161 | Species × Sites | $3.76 \times 10^{-1}$ | $3.76 \times 10^{-1}$ | 0.239 |
| Residuals | $2.09 \times 10^{-1}$ | $2.61 \times 10^{-2}$ | | Residuals | 12.60 | 1.57 | |
| C16:1 | | | | PUFA | | | |
| Species | $2.98 \times 10^{-4}$ | $2.98 \times 10^{-4}$ | 3.516 | Species | 22.40 | 22.40 | 0.554 |
| Sites | $3.76 \times 10^{-4}$ | $3.76 \times 10^{-4}$ | 4.435 | Sites | 189.00 | 189.00 | 4.666 |
| Species × Sites | $3.74 \times 10^{-5}$ | $3.74 \times 10^{-5}$ | 0.442 | Species × Sites | 1.70 | 1.70 | 0.043 |
| Residuals | $6.77 \times 10^{-4}$ | $8.47 \times 10^{-5}$ | | Residuals | 324.00 | 40.50 | |

**Table 2.** *Cont.*

| Source | Sum of Square | Mean Square | *F* Value | Source | Sum of Square | Mean Square | *F* Value |
|---|---|---|---|---|---|---|---|
| C17:1 | | | | | | | |
| Species | $1.14 \times 10^{-3}$ | $1.14 \times 10^{-3}$ | 607.382 *** | | | | |
| Sites | $8.20 \times 10^{-6}$ | $8.20 \times 10^{-6}$ | 4.375 | | | | |
| Species × Sites | $8.20 \times 10^{-6}$ | $8.20 \times 10^{-6}$ | 4.375 | | | | |
| Residuals | $1.50 \times 10^{-5}$ | $1.90 \times 10^{-6}$ | | | | | |

Note: The data of two cultivation sites (Jiulong and Leibo) of two species (iron walnut and hybrids) were used in two-way ANOVA. "*", "**", and "***" indicate significant differences ($p < 0.05$, 0.01, and 0.001) between species/sites, respectively. The same as below.

### 3.2. Micronutrient and Secondary Metabolites Levels

The contents of tocopherol, polyphenols, and flavone in walnut oil are shown in Figure 3. Three forms ($\alpha$, $\gamma$, and $\delta$) of tocopherol were detected. The total tocopherol values ranged from 109.55 (RXC) to 428.81 mg/kg (SRLD). The results clearly indicated that the predominant tocopherol form of walnut oil of iron walnut, common walnut, and the hybrid walnut trees was $\gamma$-tocopherol (86.11–350.89 mg/kg), with a proportion of more than 70%. The $\gamma$- and $\delta$-tocopherol contents of the hybrid walnut oil were significantly higher than that of common and iron walnut oil while $\alpha$-tocopherol was only detected in the iron and common walnut oil. The results of the correlation analysis showed a significant positive correlation among the contents of $\delta$-tocopherol, $\gamma$-tocopherol, and tocopherol (Figure 2). We also studied the effects of species, climatic factors, and genotype–environment interaction on the micronutrient and secondary metabolite contents based on the design of two cultivation sites (Jiulong and Leibo) of two species (iron walnut and hybrids). The results of the two-way ANOVA showed that only the species effect was significant, which indicated that genetic determinism may be the main factor affecting the tocopherol content in walnut oil (Table 3). The flavone content ranged from 0.60–5.78 mg/kg and showed similar trends to tocopherol (Figure 3). The content of flavone was not influenced by the genotypic variation or cultivation sites alone but was affected by the interaction of species × sites (Table 3). The content of polyphenols was significantly influenced by the cultivation sites and the interaction of species × sites and showed opposite trends to tocopherol and flavone (Figure 3, Table 3). The polyphenols content of the common walnut oil was significantly higher than that of the hybrid walnut oil ($F = 17.63$, $p < 0.001$), and 1.2 times higher than that of the iron walnut oil ($F = 1.187$, $p > 0.05$) (Figure 3).

**Table 3.** Two-way analysis of variance (two-way ANOVA) of the micronutrients and secondary metabolites.

| Source | Sum of Square | Mean Square | *F* Value |
|---|---|---|---|
| $\alpha$-tocopherol | | | |
| Species | 47,926 | 47,926 | 9.432 * |
| Sites | 1561 | 1561 | 0.307 |
| Species × Sites | 16,301 | 16,301 | 3.208 |
| Residuals | 40,648 | 5081 | |
| $\delta$-tocopherol | | | |
| Species | 3029.9 | 3029.9 | 18.279 ** |
| Sites | 344.5 | 344.5 | 2.078 |
| Species × Sites | 300.8 | 300.8 | 1.815 |
| Residuals | 1326.1 | 165.7625 | |
| Tocopherol | | | |
| Species | 61,329 | 61,329 | 8.943 * |
| Sites | 219 | 219 | 0.032 |
| Species × Sites | 22,853 | 22,853 | 3.332 |
| Residuals | 54,864 | 6858 | |

**Table 3.** *Cont.*

| Source | Sum of Square | Mean Square | *F* Value |
|---|---|---|---|
| Flavone | | | |
| Species | 0.0312 | 0.0312 | 0.419 |
| Sites | 0.2115 | 0.2115 | 2.842 |
| Species × Sites | 1.6337 | 1.6337 | 21.950 ** |
| Residuals | 0.5958 | 0.0744 | |
| Polyphenols | | | |
| Species | 182.1 | 182.1 | 2.823 |
| Sites | 745.1 | 745.1 | 11.550 ** |
| Species × Sites | 1176.7 | 1176.7 | 18.239 ** |
| Residuals | 516.1 | 64.5 | |

"*" and "**" indicate significant differences ($p < 0.05$ and 0.01) between species/sites, respectively.

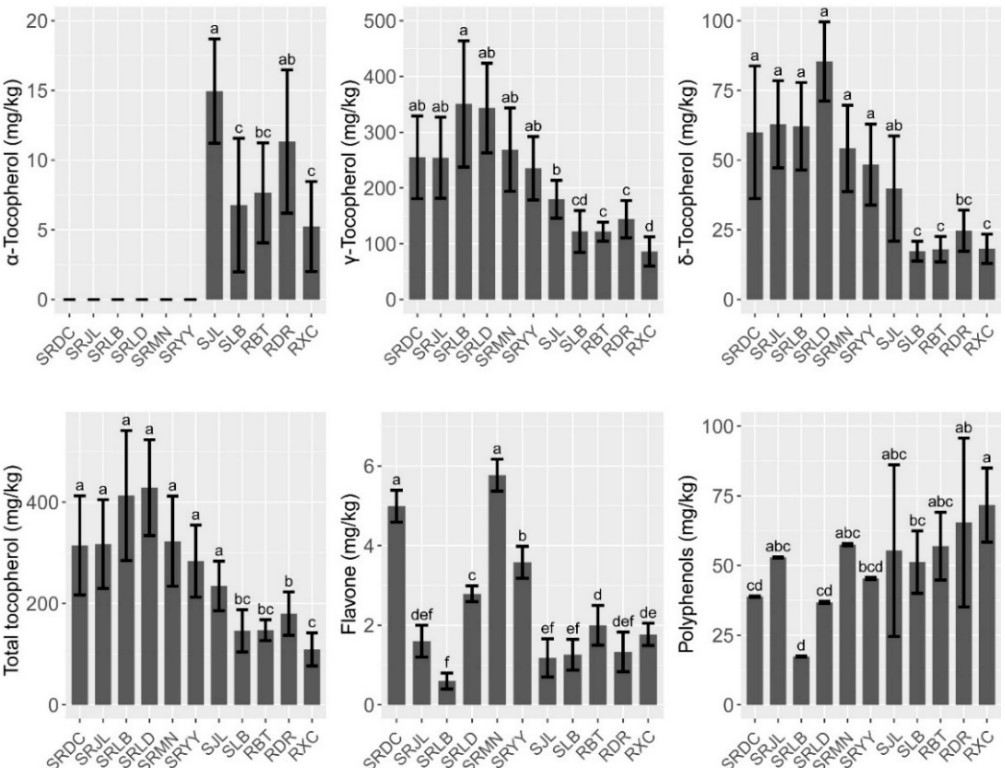

**Figure 3.** Micronutrient and secondary metabolites composition and contents of walnut oils. different letters indicate significant differences ($p < 0.05$) between different species/sites.

### 3.3. Correlation Analysis

The results of correlation between sites/species also showed that high level of correlations were found between the same species, even though cultivated in different sites (Figure 4). For instance, a highly relationship existed between the cultivated site of Leibo and Jiulong with the same species (SRLB vs. SRJL), yet a lowly relationship existed between *J. sigillata* and *J. sigillata* × *J. regia* which cultivated in the same site (SRLB vs. SLB). Similarly results also occurred in the cultivated site of Jiulong, the relationship between SRJL and SJL was lower than SRJL vs. SRMN, SRJL vs. SRDC, and SRJL vs. SRYY.

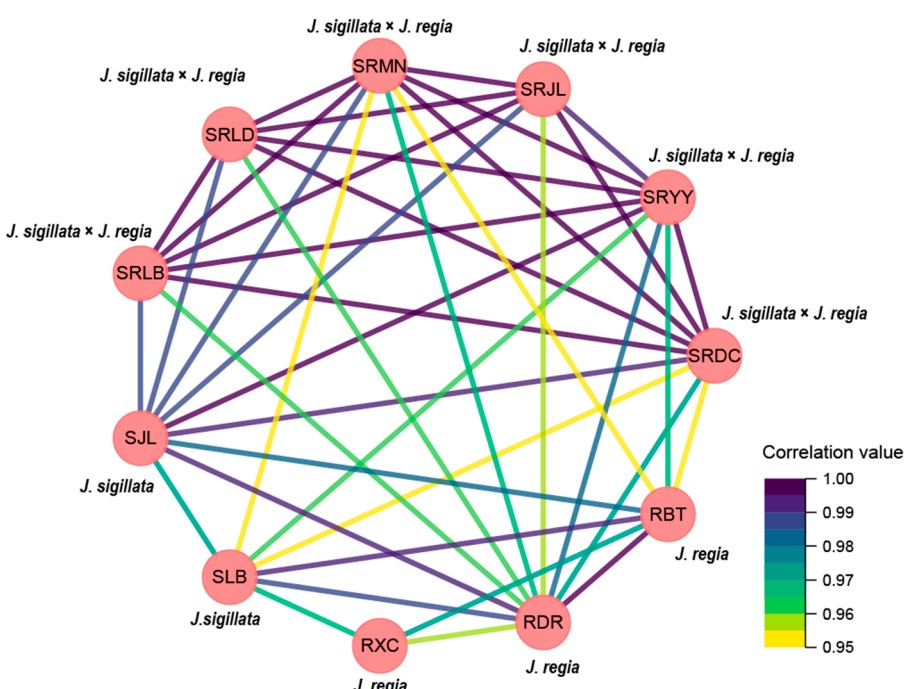

**Figure 4.** The related network diagram of cultivation sites/species and the main composition of walnut oil.

### 3.4. Climatic Factors and Fatty Acid Composition

The effects of climatic factors on the fatty acid composition throughout the fruit development period from the end of flowering to physiological maturity were analyzed. Precipitation showed a significant positive correlation with myristic acid (C14:0), palmitic acid (C16:0), margaric acid (C17:0), and SFAs, and a significant negative correlation with eicosenoic acid and polyphenols (Figure 5). Minimum relative humidity (MinRH) and average relative humidity (AvgRH) showed a significant negative correlation with oleic acid (C18:1) and MUFAs. Nevertheless, MinRH had a positive significant effect on linoleic acid (C18:3) and PUFA. There was also significant negative correlation between palmitic acid (C16:0), margaric acid (C17:0), and margaroleic acid (C17:1) and between average relative humidity (AvgRH) and oleic acid (C18:1) and MUFAs. Further, after multiple regression model comparison and averaging, the model explained 85%, 91%, and 65% of the variation in oleic acid (C18:1), linoleic acid (C18:3), and linolenic (C18:2), respectively. The average relative humidity (AvgRH) explained the most variation in both oleic (C18:1) (positive) and linolenic acid (C18:2) (negative) (Figure 6). Latitude explained the most in linoleic acid (C18:3), and other variables had a significant but small effect on linoleic acid (C18:3). The model explained 83%, 85%, and 97% of the variation in PUFAs, MUFAs, and SFAs, respectively, The AvgRH also explained the most variation in PUFAs (negative), MUFAs (positive), and SFAs (positive). Other variables with a small effect included temperature, precipitation, altitude, etc. Regarding the micronutrients and secondary metabolites, AvgRH was most important (negative) in tocopherol and flavone, and latitude explained the most variation (positive) in polyphenols. The model explained 52% and 61% of the variation in polyphenols and flavone, respectively. Other variables had a small effect on them, e.g., the estimated value of the minimum temperature (MinTem) on flavone was 11.33, which is lower than the absolute estimated value of AvgRH (193.62).

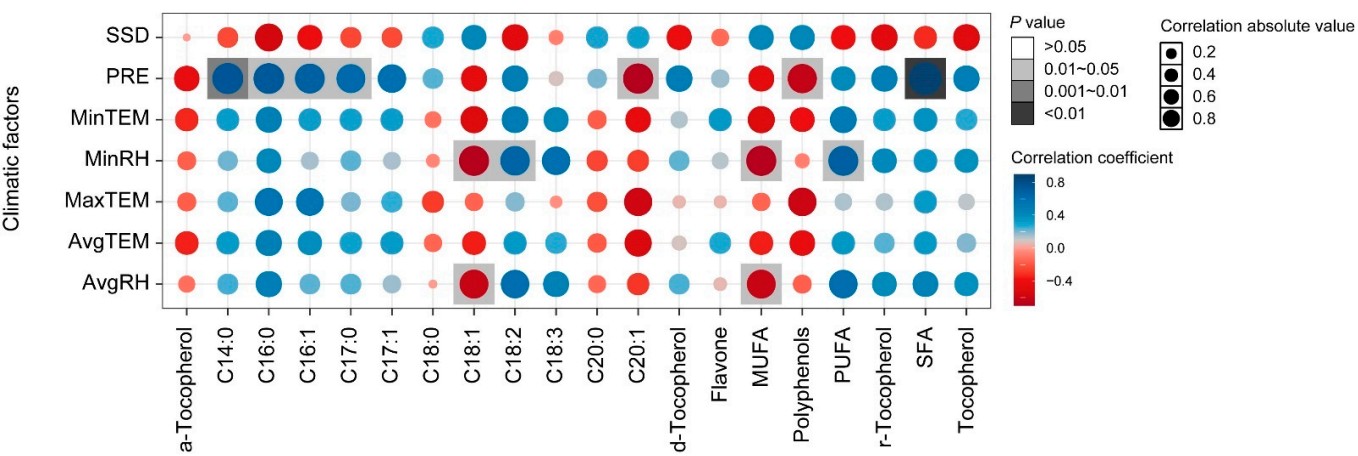

Chemical composition of walnut oil

**Figure 5.** Correlation analyses of the climatic factors and chemical composition of walnut oil. Note: MinRH, minimum relative humidity (%); AvgRH, average relative humidity (%); AvgTEM, average temperature (°C); MaxTEM, maximum temperature (°C); MinTEM, minimum temperature (°C); PRE, precipitation (mm); SSD, sunshine duration (h); the same as below.

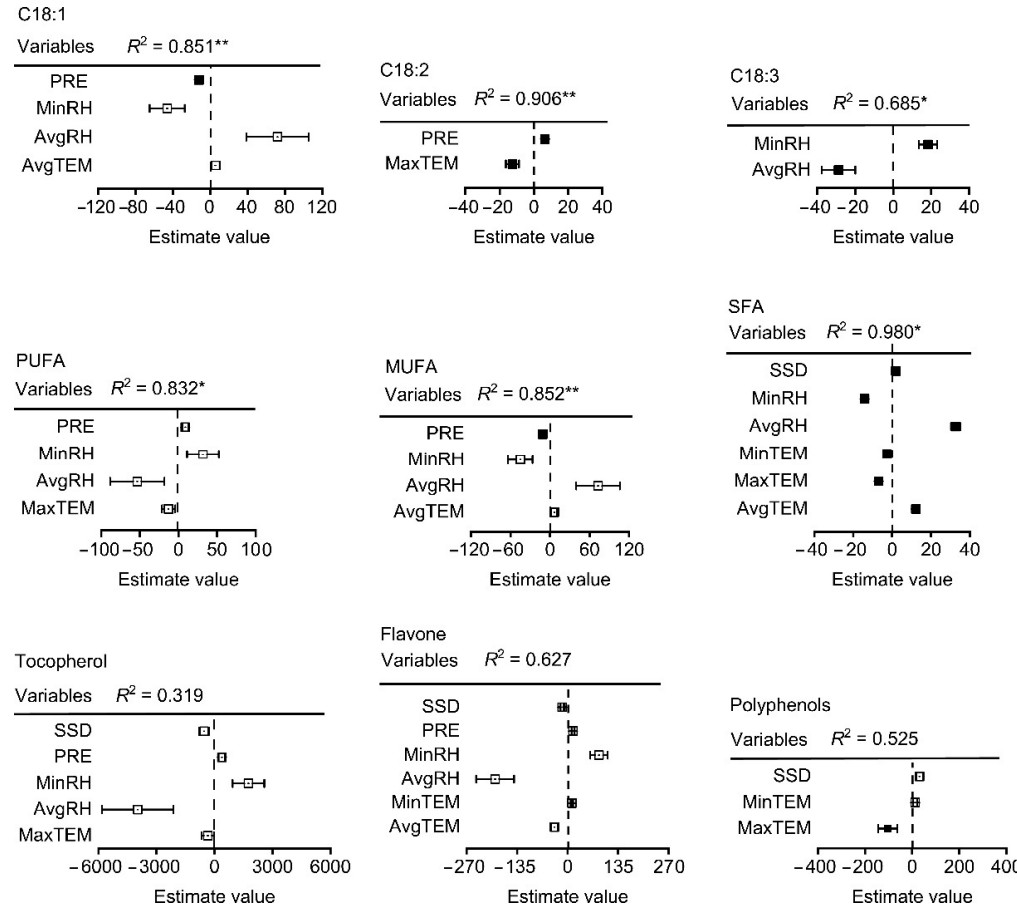

**Figure 6.** Results of multiple regressions after the selection process of the climatic factors affecting the fatty acid content of walnut oil. Note: Each variable was standardized before comparing effect sizes (squares) to determine differences in the strength of the predictor variables. Variables were selected based on a full set regression. Closed squares indicate significant effects ($p < 0.05$) and lines indicate standard errors. * and ** indicate a significant difference ($p < 0.05$, 0.01), respectively.

## 4. Discussion

The current results present the fatty acid, micronutrients, and secondary metabolites analysis carried out on different species of walnut and as such are important for determining the nutritional value and potential uses of walnut oil. Similar to other results reported in *Juglans sigillata* Dode and *Juglans regia* L., the walnut oil of *Juglans sigillata* Dode, *Juglans regia* L., and *Juglans sigillata* × *regia* L. was mainly composed of palmitic, stearic, and oleic acid, etc. [2]. In the current study, there were significant differences in the fatty acid composition and content between species/genotypes. Consistent with the results of our study, the fatty acid content in some vegetable oils was significantly different between cultivars, varieties, and species, such as walnut and almond oil [2,5,6,9]. Gao et al. [1] also reported that the species of walnut was an important factor affecting the fatty acid composition of walnut oil [1]. The results of the comparative and relationship analysis also indicated that genetic differences also play an important role in the regulation of the compositions and contents of walnut oil, especially at the level of species. Walnut oil from the same species mostly contained the same chemical compositions (Figure 1). The interaction of gene differences among different genotypes resulted in differences in the content of lipid components [22]. Hence, walnut species and genotypes may affect the fatty acid composition and content in walnut oil. In three species, a significant negative correlation was identified between the oleic and linoleic acid content, which was also found in *Olea europaea*, *Torreya grandis*, *J. regia*, and *J. sigillata* oil [2,6,23,24]. The contents of oleic and linoleic acid have a relative balance, because in the accumulation progress of fatty acid in walnut kernels, there are two pathways in the production of oleoyl-ACP: hydrolysis to produce oleic acid and ACP, and further dehydrogenation to produce linoleoyl-ACP (C18:2-ACP) [24]. There were also interspecific differences in the composition of minor fatty acids, for instance, C14:0, C17:0, and C17:1 were only detected in the hybrid walnut oil. Similar results were also reported in common walnut oil from northeastern Italy and northwestern, southwestern, and eastern China [3,6]. However, C17:0 has previously been detected in common and iron walnut oil from Yunnan Province in China and northeast Portugal [2,5], and C17:1 has previously been detected in common walnut (*J. regia*) oil [5]. Gao et al. (2019) reported that C16:1 was not detected in common walnut. However, C16:1 has previously been detected in common walnut oil [25] and in this study. The levels of gene expression related to fatty acid biosynthesis are regulated by the growth environment during the process of fruit development [26]. Therefore, we hypothesized that climatic factors would play an important role in the fatty acid composition of walnut oil. Furthermore, the results of the two-ANOVA showed that the composition and content of fatty acid were significantly influenced by the climate of the cultivation sites and interspecific genetic differences. Crews et al. [12] and Gao et al. [6] reported similar results in that differences in the geographic environment led to differences in the fatty acid composition and content. Therefore, the climate of the cultivation area of walnut may the main factor affecting the composition and content of fatty acid in walnut oil, especially for minor fatty acids.

Tocopherol is a term used to refer to a group of minor but ubiquitous lipid-soluble compounds [27]. The γ-tocopherol content of iron walnut oil was significantly higher than that of common walnut oil, whereas Gao et al. [2] found that common walnut oil provided higher tocopherols than iron walnut oil. The cultivars also affect the tocopherol content of the walnut oil [2,28]. After combining the analysis of the two-way ANOVA of α-tocopherol, δ-tocopherol, and tocopherol, we speculated that genotypic variation was the main factor that led to the differences in the tocopherol contents between the three *Juglans* species. Polyphenols can inhibit the oxidation activity of low-density lipoprotein, and the polyphenols of walnut oil have significant effects as antioxidants [7,29]. It was found that the polyphenols content of the iron walnut in our study was higher than that in Yunnan Province, China [2], but similar to Xinjiang Uygur Autonomous Region, China [1]. Therefore, the planting conditions may affect the polyphenols content in walnut oil in China. A similar conclusion can be advanced in the case of the two-way ANOVA of polyphenols; the species effect was not significant, but the significant site and species × site

interaction shows that the content of polyphenols is mainly affected by the cultivation climatic conditions depending on the species.

Environmental factors, such as temperature, drought, and latitude, may affect the oil composition of most oil crops [3]. A number of recent studies indicated that temperature was one of the main factors affecting the oil composition of oil crops such as sunflower [15], soybean [16], walnut [3], and almond oil [10]. Poggetti et al. (2018) observed a significant positive correlation between the daily minimum temperature and oleic acid content in wild walnuts (*J. regia* L.). This differs from the findings presented here showing that no significant relationship was found between any fatty acid content and temperature (average temperature, maximum temperature, and minimum temperature) and sunshine duration. These results indicated that the temperature and sunshine duration at the cultivation sites were not the main climatic factors affecting the chemical composition of walnut oil. A possible explanation for this might be that the level of genetic differences among the species and its effect on the fatty acid composition and contents was higher than daily temperature. The relative humidity regulated the MUFA and PUFA contents, which control the content of oleic and linoleic acid, evidencing a similar effect model and positive/negative significant correlation between relative humidity and oleic acid and linoleic acid. Furthermore, for most of the fatty acids, micronutrients, and secondary metabolites in walnut oil, relative humidity was the most important factor among the climatic factors. Therefore, the relative humidity was the key climatic factor in the assessment of the adaptable varieties at the cultivation sites, especially for the selection of the species level. The mechanisms through which relative humidity influences MUFAs and PUFAs deserve further research, given that knowledge of the climatic factors affecting fatty acid synthesis, accumulation, and material transformation is helpful for elite germplasm selection and new breeding genotype development [3].

## 5. Conclusions

These findings clearly indicate that walnut oil from different species can be distinguished by their fatty acid compositions, micronutrients, and secondary metabolites. Common walnut had a lower PUFA content but higher MUFA content than iron and hybrid walnut. Hybrids walnut oil contained C14:0, C17:0, and C17:1, which was not detected in common and iron walnut oil. In addition, the contents of micronutrients and secondary metabolites of the hybrid walnut oil were higher than those of common and iron walnut oil, but α-tocopherol was only detected in common and iron walnut oil. These findings could be used to help walnut breeding and for efficient utilization of walnut oil. Genetic and climatic factors and their interaction emerged as reliable predictors of the fatty acid, micronutrient, and secondary metabolite compositions and contents of walnut oil, especially interspecies genetic differences. Climatic factors play an important role in walnut oil quality, and relative humidity, mainly average relative humidity (AvgRH), was the most important factor regulating the composition and content of walnut oil. Overall, this study strengthens the idea that interspecies genetic differences were the chief genetic factor, and relative humidity was the chief climatic factor for matching sites with walnut trees, which could contribute to the obtainment of raw materials of walnut kernels with high values of oil nutrients.

**Author Contributions:** H.Y., X.W. and X.X. participated in the design of the study and performed the statistical analysis, H.Y., X.X., J.L. and F.W. helped to collect the data, X.X., J.L., J.M. and Y.S. helped to determine the index, H.Y. and L.C. drafted the manuscript, F.H. and F.Z. helped to revise the manuscript. All authors have read and agreed to the published version of the manuscript.

**Funding:** This work was supported by Key Research and Development Project of Sichuan Province (2021YFYZ003), and the Science and Technology Project of Sichuan Province (2020YFN0058).

**Institutional Review Board Statement:** Not applicable.

**Informed Consent Statement:** Not applicable.

**Conflicts of Interest:** The authors declare no potential conflict of interest.

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
