# Peer review of "Chemical Compositions of Walnut (Juglans Spp.) Oil: Combined Effects of Genetic and Climatic Factors"

_forests, doi:10.3390/f13060962_

Round 1
Reviewer 1 Report
There is a problem regarding considering Juglans regia and J. sigillata as different species, since in the latest phylogenetic works (2017, 2021) they seem to be the same species.
It is stated that the 10 samples that were examined are different genotypes. How were the genotypes evaluated and characterized? Why only two samples of J. regia and two of J. sigillata and six of hybrids?
China is one of the main producers of walnuts and the distribution area of ​​Juglans regia is wide, so it is considered that the environmental conditions can be widely variable, and it is important to explain how the sites were chosen if contrasting conditions were sought or not. It is also important to explain how the number of sites was defined.
The environmental characterization depends on many elements, both natural and cultural, some of which influence the production, quality, concentration, and composition of the different metabolites, among the most important the climate (temperature, precipitation, wind flow, humidity, evaporation, etc.) and the soil (type, composition, permeability, etc.) How were the sites characterized? What are the main differences between them? Why soil variables were not included, since it is known that they can explain some differences between the production of some metabolites? There are not phenological variables among the considered taxa? orography?
It is necessary to discuss the results widely, considering experiences in different species and different countries, trying to relate the variation in the production of metabolites and environmental variables, discussing other variables that can explain the results such as physiological and phenological, if relevant.
Author Response
List of Responses
Dear Reviewer:
Thank you for your letter and for the reviewers’ comments concerning our manuscript entitled “Chemical compositions in walnut (Juglans spp.) oil: combined effects of genetic and environmental factors” (Manuscript ID: forests-1745433). Those comments are all valuable and very helpful for revising and improving our paper, as well as the important guiding significance to our researches. We have studied comments carefully and have made correction which we hope meet with approval. Revised portion are marked in “Track Changes” in the paper. The main corrections in the paper and the responds to yours comments are as following:
- There is a problem regarding considering Juglans regiaand sigillata as different species, since in the latest phylogenetic works (2017, 2021) they seem to be the same species.
Response: Although some reports consider that Juglans sigillata is a landrace or ecotype of J. regia (Zhang et al, 2019; Yuan et al, 2018), the genome-wide phylogenetic tree of single-copy orthologous genes indicated an estimated divergence time between J. sigillata and J. regia of 49 million years ago, which support that J. regia is a sister member of J. sigillata in section Dioscaryon Dode (Ning et al, 2020). Wang et al (2008) suggested that J. sigillata may be a sub-species of J. regia basis on the genetic diversity results of SSR marker. Therefore, the systematic status of J. sigillata in Juglans remains controversial. In our opinion, (1) J. sigillata only distributed in China, and the most authority taxonomy reference, Flora of China, considered J. sigillata and J. regia separated as two independent species (Flora of China, 1999, 4: 282-293). (2) There were significant morphological differences between J. sigillata and J. regia, especially in nut and leaf. (3) The distribution of J. regia and J. sigillata do not overlap. J. regia with a distribution from southeastern Europe to China and the Himalayas, and J. sigillata distributed in Yunnan, Guizhou, Sichuan, and Tibet of Southwest China. (4) There are also significant differences of the habitat conditions between the distribution of J. regia and J. sigillata. Based on the above considerations, we considering J. regia and J. sigillata as different species.
Reference:
Zhang BW, Xu LL, Li N, et al. Phylogenomics reveals an ancient hybrid origin of the Persian walnut. Molecular Biology and Evolution, 2019, 26 (11): 2451-2461.
Yuan XY, Sun YW, Bai XR, et al. Population structure, genetic diversity, and gene introgression of two closely related walnuts (Juglans regia and J. sigillata) in southwestern China revealed by EST-SSR markers. Forests, 2018, 9, 646.
Ning DL, Wu T, Xiao LJ, et al. Chromosomal-level assembly of Juglans sigillata genome using Nanopore, BioNano, and Hi-C analysis. GigaScience, 2020, 9: 1-9.
Wang H, Pei D, Gu RS, et al. Genetic diversity and structure of walnut populations in central and southwestern China revealed by microsatellite markers. J. Am. Soc. Hortic. Sci, 2008, 133: 197-203.
- It is stated that the 10 samples that were examined are different genotypes. How were the genotypes evaluated and characterized? Why only two samples of regiaand two of J. sigillata and six of hybrids? China is one of the main producers of walnuts and the distribution area of Juglans regia is wide, so it is considered that the environmental conditions can be widely variable, and it is important to explain how the sites were chosen if contrasting conditions were sought or not. It is also important to explain how the number of sites was defined.
Response: 11 samples (three J. regia samples in Batang, Xiangcheng, and Derong, two J. sigillata samples in Jiulong and Leibo, six hybrids samples in Jiulong, Leibo, Dechang, Luding, Mianning, and Yanyuan) were collected and examined to evaluate the genetic and environmental effect. J. sigillata and hybrids were cultivated in the same site of Leibo, Liangshan, China and Jiulong, Ganzi, China, which we could evaluated the effect of genotype variation on walnut oil. As reviewer pointed out that environmental conditions can be widely variable with the wide distribution of walnuts. ‘Yanyuanzao’ is a commercial cultivar through crossbreeding of J. sigillata × J. regia. Hybrids (‘Yanyuanzao’) is the most widely cultivated genotype under Hengduan Mountains, China, six samples (the main distribution area of ‘Yanyuanzao’) were collected to evaluate the effect of environmental factors on walnut oil in ‘Yanyuanzao’. For better compare the effects of genetic and environmental factors on walnut oil, as the hybrid parents of ‘Yanyuanzao’, two J. sigillata and three J. regia samples (the main distribution area under Hengduan Mountain) were collected to evaluate the environmental factors on walnut oil. The samples of J. regia and J. sigillata located in the main distribution of hybrids, which is also the main distribution area of J. regia and J. sigillata under the cultivated distribution of hybrids. Thus, such sampling (two samples of J. sigillata, three samples of J. regia, and six samples of hybrids) would make the results more representative and meaningful, and provide more interpretation for the effect of genetic and environmental factors on walnut oil.
- The environmental characterization depends on many elements, both natural and cultural, some of which influence the production, quality, concentration, and composition of the different metabolites, among the most important the climate (temperature, precipitation, wind flow, humidity, evaporation, etc.) and the soil (type, composition, permeability, etc.) How were the sites characterized? What are the main differences between them? Why soil variables were not included, since it is known that they can explain some differences between the production of some metabolites? There are not phenological variables among the considered taxa? orography?
Response: As reviewer suggested that the climate and soil factors significantly affected on the production, quality, concentration, and composition of the walnut. In this paper, we explore the effect of climatic factors on the quality of walnut oil. Because the latitude and meteorological factors are the main differences between cultivation areas. The tree management and soil nutrient and moisture management in our sampled cultivation areas are similar, such as fertilize, trim, and pest control, etc. Unfortunately, we could not obtain the phenological data in these sampled cultivation areas and species, because it is very difficult to investigate the phenological in the nine sites simultaneously. Considering the reviewer’s suggestion, we have revised the title as chemical compositions in walnut (Juglans spp.) oil: combined effects of genetic and environmental climatic factors), and related content in main text of this manuscript.
- It is necessary to discuss the results widely, considering experiences in different species and different countries, trying to relate the variation in the production of metabolites and environmental variables, discussing other variables that can explain the results such as physiological and phenological, if relevant.
Response: Considering the reviewer’s suggestion, we have supplemented the correlation analysis between variation in the production of walnut oil components and climatic variables. And deepened the discussion according to the results of correlation analysis. Please review line line225-234, Figure 5, and line327-331. We also deepen the discussion section, please review line276-279, line291-295, line308-311, and line327-331.
Special thanks to you for your good comments.
We appreciate for your warm work earnestly, and hope that the correction will meet with approval.
Once again, thank you very much for your comments and suggestions.

Reviewer 2 Report
The paper presents the main environmental factors analysis affecting oil composition and content of Walnut oil determined in three species: Juglans regia L. (common walnut), Juglans sigillata Dode (iron walnut), and their hybrids (Juglans sigillata Dode×Juglans regia L.) cultivated in different sites. The aim of the studies was (1) to distinguish walnut oil from different species/genotypes, (2) to determine the species, cultivated site, and the interaction species × site effect on walnut oil, and (3) to evaluate the effect of the main environmental factors during nut growth on fatty acid.
The fatty acid compositions and micronutrient contents in walnut oil were examined. The major fatty acids were linoleic, and linolenic acid. High variation in fatty acid and micronutrient contents were both found between species and sites. The myristic, margaric, and margaroleic acid was only detected in hybrids walnut oil, yet α-tocopherol was only detected in common and iron walnut oil. Environment factors significantly affected the composition and content of fatty acids, whereas δ-tocopherol, was mostly dependent on genetics. Average relative humidity explained the most variation in fatty acids and micronutrients, which showed a significantly positive and negative effect on monounsaturated fatty acids and polyunsaturated fatty acids, respectively.
The paper possesses potential practical guide value to contribute to better the matching sites with walnut trees, and nutritional value improvement of walnut oil. The paper could be rewritten to be more clear to the audience. This practical value of the paper related to interactions genotype × site variation could be emphasized in studies.
The presentation of the results could be improved by the authors to be more understandable to the readers. It is very difficult to follow the results through the tables, maybe the good advice is to show some data in figures, while in tables to present only statistical test results. Especially the problem touches table 2. There is some problem with fonts e.g. Celsius degree; line 100-103; oC should superscript oC.
Author Response
List of Responses
Dear Reviewer:
Thank you for your letter and for the reviewers’ comments concerning our manuscript entitled “Chemical compositions in walnut (Juglans spp.) oil: combined effects of genetic and environmental factors” (Manuscript ID: forests-1745433). Those comments are all valuable and very helpful for revising and improving our paper, as well as the important guiding significance to our researches. We have studied comments carefully and have made correction which we hope meet with approval. Revised portion are marked in “Track Changes” in the paper. The main corrections in the paper and the responds to the yours comments are as following:
- The paper could be rewritten to be more clear to the audience. This practical value of the paper related to interactions genotype × site variation could be emphasized in studies.
Response: We have revised the full manuscript carefully for readability according to the reviewer’s suggestion. And we also emphasized the results of interactions genotype × site in the part of results (line179-181) and discussion (line295-298).
- The presentation of the results could be improved by the authors to be more understandable to the readers. It is very difficult to follow the results through the tables, maybe the good advice is to show some data in figures, while in tables to present only statistical test results. Especially the problem touches table 2. There is some problem with fonts e.g. Celsius degree; line 100-103; oC should superscript o
Response: Considering the reviewer’s suggestion, we have replaced the data in table 2 by figures. Please review figure 1 and figure 3. We also have revised the fonts. Please review line 103-107.
Special thanks to you for your good comments.
We tried our best to improve the manuscript and made some changes in the manuscript. We appreciate for yours warm word earnestly, and hope that the correction will meet with approval.
Once again, thank you very much for your comments and suggestions.

Reviewer 3 Report
Dear Authors,
I attach a review of the article „Chemical compositions in walnut (Juglans spp.) oil: combined effects of genetic and environmental factors”.
The aim of this work was:
(1) to distinguish walnut oil from different species/genotypes,
(2) to determine the species, cultivated site, and the interaction species × site effect on walnut oil,
(3) to evaluated effect of the main environmental factors during nut development on fatty acid.
General comment:
The (2) purpose of the research was not achieved.
The use of two-way ANOVA is possible in the following situation:
3 species/genotypes cultivated in each of the 9 sites = 27 plots.
However is situation when: Line 83-86 - … The commercial cultivar was ‘Yanyuanzao’ (a hybrid of J. sigillata Dode×Juglans regia L., widely cultivated in China) which cultivated in six sites under Hengduan Mountains. J. regia was cultivated in three sites, and J. sigillata Dode was cultivated in two sites. …
(1 species/genotype x 6 sites) + (1 species/genotype x 3 sites) (1 species/genotype x 2 sites) = 11 plots.
Other comments:
Statistical analysis
Line 125-127: … Statistical analysis was performed using R version 4.0.5, utilizing the homogeneity of variance and normal distribution test, Shapiro-Wilk normality, one-way ANOVA, and two-way ANOVA test to analyze the normality condition. …
Rev: very complicated and unclear, two-way ANOVA test to analyze the normality condition?
Rev: what test was used to testing the homogeneity?
Line 128-129: …Correlation analysis between chemical characteristics were carried out by Pearson’s test. …
Rev: maybe Pearson correlation coefficient?
p-value should be given as p<0.05
Results
Line 142-144: … The content of PUFAs (Shapiro-Wilk normality test. W = 142 31.624, P = 0.000), linoleic (W = 29.785, P = 0.001), and linolenic acid (W = 33.569, P = 0.000) 143 were significantly different in different walnut species/genotypes. …
Rev: presenting the results of the Shapiro-Wilk test is unnecessary - it causes confusion
Line 147-148: … and slightly higher than iron walnut (J. sigillata) (F = 1.785, P = 0.191), …
Rev: it can't be slightly higher; results – F = 1.785, p = 0.191 clearly shows that is not higher
Line 151-152: … Notably, there was a negative correlation between the content of oleic and linoleic acid (r = -0.979, P = 0.000). …
Rev: correlation results should be presented in a table
Line 167-170: … The γ- and δ-tocopherol content of hybrids walnut oil were significantly higher than that of common (Wγ-tocopherol = 29.034, P = 0.000; Wδ-tocopherol = 29.584, P = 0.000) and iron walnut oil (Fδ-tocopherol = 33.14, P = 0.000; Fδ-tocopherol = 29.41, P = 0.000), while the α-tocopherol was only detected in iron and common walnut oil. …
Rev: presenting the results of the Shapiro-Wilk test is unnecessary - it causes confusion
Line 180: Table2
Rev: the table format is not acceptable
Line 181 … Note: …
Rev: the information about the analysis of variance that has been performed should appear here
Line 183-184: ….Table 3. Two-way analysis of variance (two-way ANOVA) for micronutrient. …
Rev: the title should be above the table
Line 200: Figure 1
Rev: legends should be explained
Rev: A, legend species - names should be unified
Line 200: … cultivated sites/species …
Rev: cultivated sites?
References
Line 323: … Juglans regia and Juglans sigillata …
Rev: should be in italics
Line 339: … major fatty Acid composition …
Author Response
List of Responses
Dear Reviewer:
Thank you for your letter and for the reviewers’ comments concerning our manuscript entitled “Chemical compositions in walnut (Juglans spp.) oil: combined effects of genetic and environmental factors” (Manuscript ID: forests-1745433). Those comments are all valuable and very helpful for revising and improving our paper, as well as the important guiding significance to our researches. We have studied comments carefully and have made correction which we hope meet with approval. Revised portion are marked in “Track Changes” in the paper. The main corrections in the paper and the responds to the yours comments are as following:
- The (2) purpose of the research was not achieved.
Response: Its really true as reviewer pointed out that the (2) purpose of the research would be not achieved. In this case, we want to explore the effect of genotype variation and cultivation site on chemical composition of walnut oil from different species. Two species (J. sigillata and J. sigillata × J. regia) which cultivated in the same two sites (Leibo and Jiulong) in this study were used to calculated the genetic and environmental effect. We have revised the (2) purpose, which replaced by “to determine the effect of genotypes and climatic factors on walnut oil”.
- The use of two-way ANOVA is possible in the following situation: 3 species/genotypes cultivated in each of the 9 sites = 27 plots. However is situation when: Line 83-86 - … The commercial cultivar was ‘Yanyuanzao’ (a hybrid of J. sigillata Dode×Juglans regia L., widely cultivated in China) which cultivated in six sites under Hengduan Mountains. J. regia was cultivated in three sites, and J. sigillata Dode was cultivated in two sites. …(1 species/genotype x 6 sites) + (1 species/genotype x 3 sites) (1 species/genotype x 2 sites) = 11 plots.
Response: Three species/genotypes were selected in our study. Six hybrids cultivated in six different sites. Three J. regia cultivated in three different sites. Two J. sigillata cultivated in two different sites. In our experimental design, two J. sigillata and two hybrids cultivated in the same two sites (Jiulong and Leibo). Therefore, the data of 2 species × 2 sites was used in two-way ANOVA (two factors: 2 species and 2 sites).
Line 125-127: … Statistical analysis was performed using R version 4.0.5, utilizing the homogeneity of variance and normal distribution test, Shapiro-Wilk normality, one-way ANOVA, and two-way ANOVA test to analyze the normality condition. …
Rev: very complicated and unclear, two-way ANOVA test to analyze the normality condition?
Rev: what test was used to testing the homogeneity?
Response: We very sorry for our incorrect writing, and we have rewritten this part. The functions of shapiro. test and bartlett. test were used to calculated the normal distribution and homogeneity of variance test. And then, one-way and two-way ANOVA test were used to determine the different of chemical composition of walnut oil between different species and cultivation sites. Please review line 130-134.
- Line 128-129: …Correlation analysis between chemical characteristics were carried out by Pearson’s test. …
Rev: maybe Pearson correlation coefficient?
p-value should be given as p<0.05
Response: Considering the reviewer’s suggestion, we have revised this part as “Pearson correlation coefficient between chemical characteristics were calculated (α = 0.05)”
- Line 142-144: … The content of PUFAs (Shapiro-Wilk normality test. W = 142 31.624, P = 0.000), linoleic (W = 29.785, P = 0.001), and linolenic acid (W = 33.569, P = 0.000) 143 were significantly different in different walnut species/genotypes. …
Rev: presenting the results of the Shapiro-Wilk test is unnecessary - it causes confusion
Response: We have deleted the results of Shapiro-Wilk test in the manuscript according to the reviewer’s suggestion.
- Line 147-148: … and slightly higher than iron walnut (J. sigillata) (F = 1.785, P = 0.191), …
Rev: it can't be slightly higher; results – F = 1.785, p = 0.191 clearly shows that is not higher
Response: It is really true as reviewer suggested that the results of the content of PUFAs showed not significant difference between hybrids and iron walnut oil. There was not higher content of PUFAs in hybrids than that in iron walnut. We have deleted this content in the sentence.
Line 151-152: … Notably, there was a negative correlation between the content of oleic and linoleic acid (r = -0.979, P = 0.000). …
Rev: correlation results should be presented in a table
Response: As reviewer suggested that we have supplemented the correlation results with figure. Please review figure 2.
- Line 167-170: … The γ- and δ-tocopherol content of hybrids walnut oil were significantly higher than that of common (Wγ-tocopherol = 29.034, P = 0.000; Wδ-tocopherol = 29.584, P = 0.000) and iron walnut oil (Fδ-tocopherol = 33.14, P = 0.000; Fδ-tocopherol = 29.41, P = 0.000), while the α-tocopherol was only detected in iron and common walnut oil. …
Rev: presenting the results of the Shapiro-Wilk test is unnecessary - it causes confusion
Response: We have deleted the results of Shapiro-Wilk test in the manuscript according to the reviewer’s suggestion.
- Line 180: Table2
Rev: the table format is not acceptable
Response: Considering the reviewer’s suggestion, we have replaced the data in table 2 by figures. Please review figure 1 and figure 3.
- Line 181 … Note: …
Rev: the information about the analysis of variance that has been performed should appear here
Response: Considering the reviewer’s suggestion, we have supplemented the analysis of variance of polyphenols content between common and iron walnut oil, and supplemented the F and P value in this sentence.
- Line 183-184: ….Table 3. Two-way analysis of variance (two-way ANOVA) for micronutrient. …
Rev: the title should be above the table
Response: We have put the title of table 3 above the table.
- Line 200: Figure 1
Rev: legends should be explained
Rev: A, legend species - names should be unified
Response: We deleted figure 4A in Figure 4 (original Figure 1), considered that there would be a repeat with Figure 1 (original Table 1) and Figure 4 (original Figure 1B). And we have unified the legend species names in Figure 4 with the manuscript.
- Line 200: … cultivated sites/species …
Rev: cultivated sites?
Response: We have revised cultivated sites as cultivation sites in the full manuscript. The related network diagram showed the correlation between cultivation sites according to the chemical composition and contents of walnut oil, and also showed the correlation between species according to the chemical composition and contents of walnut oil.
- Line 323: … Juglans regia and Juglans sigillata …
Rev: should be in italics
Response: We have revised it in bibliographic reference.
- Line 339: … major fatty Acid composition …
Response: We are very sorry for our negligence of this error, and we have revised it.
Special thanks to you for your good comments.
We tried our best to improve the manuscript and made some changes in the manuscript. We appreciate for yours warm word earnestly, and hope that the correction will meet with approval.
Once again, thank you very much for your comments and suggestions.
Reviewer 4 Report
In the present manuscript, the chemical composition of the seed oil of three walnut species was performed.
Correct "cultivated sites" to "cultivation sites" throughout the text.
Line 74: rephrase to "between genotypes, cultivation sites, and their interaction".
Table 2 should be reformatted to improve readability.
Correct the place of the legend of Table 3. How the authors performed a two-way analysis in this experimental design?
The concept of the study was to evaluate the effect of Genotype and the Environment on the chemical composition of walnut seed oil. However, according to Table 1, only iron walnut and the hybrid were cultivated at the same site.
Author Response
List of Responses
Dear Reviewer:
Thank you for your letter and for the reviewers’ comments concerning our manuscript entitled “Chemical compositions in walnut (Juglans spp.) oil: combined effects of genetic and environmental factors” (Manuscript ID: forests-1745433). Those comments are all valuable and very helpful for revising and improving our paper, as well as the important guiding significance to our researches. We have studied comments carefully and have made correction which we hope meet with approval. Revised portion are marked in “Track Changes” in the paper. The main corrections in the paper and the responds to the yours comments are as following:
- Correct "cultivated sites" to "cultivation sites" throughout the text.
Response: We have revised cultivated sites as cultivation sites in the full manuscript.
- Line 74: rephrase to "between genotypes, cultivation sites, and their interaction".
Response: Considering the reviewer’s suggestion, we have rewritten this sentence. Such as “Therefore, comprehensive analysis the difference of fatty acid composition and content in walnut oil among species/, genotypes, and cultivation sites, and their interaction is important for breeding programs and cultivation (matching site with species or varieties)”
- Table 2 should be reformatted to improve readability.
Response: Considering the reviewer’s suggestion, we have replaced the data in table 2 by figures. Please review figure 1 and figure 3.
- Correct the place of the legend of Table 3. How the authors performed a two-way analysis in this experimental design?
Response: We have put the title of table 3 above the table. In our experimental design, two J. sigillata and two hybrids cultivated in the same two sites (Jiulong and Leibo). Therefore, the data of 2 species × 2 sites was used in two-way ANOVA (two factors: 2 species and 2 sites).
- The concept of the study was to evaluate the effect of Genotype and the Environment on the chemical composition of walnut seed oil. However, according to Table 1, only iron walnut and the hybrid were cultivated at the same site.
Response: Three species/genotypes were selected in our study. Six hybrids cultivated in six different sites. Three J. regia cultivated in three different sites. Two J. sigillata cultivated in two different sites. In this case, we want to explore the effect of genotype variation and cultivation site on chemical composition of walnut oil from different species. Two species (J. sigillata and J. sigillata × J. regia) which cultivated in the same two sites (Leibo and Jiulong) in this study were used to calculated the genetic and environmental effect. We have revised the (2) purpose, which replaced by “to determine the effect of genotypes and climatic factors on walnut oil”.
Special thanks to you for your good comments.
We tried our best to improve the manuscript and made some changes in the manuscript. We appreciate for yours warm word earnestly, and hope that the correction will meet with approval.
Once again, thank you very much for your comments and suggestions.
Round 2
Reviewer 1 Report
The explanations of the authors are consistent but are not reflected in their manuscript. None of the questions and suggestions from the first review were considered. The changes that were made are fundamentally format.
Author Response
List of Responses
Dear Reviewer:
Thanks again for your comments concerning our manuscript entitled “Chemical compositions in walnut (Juglans spp.) oil: combined effects of genetic and climatic factors” (Manuscript ID: forests-1745433). Those comments are all valuable and very helpful for revising and improving our paper, as well as the important guiding significance to our researches. We have studied comments carefully and have made correction which we hope meet with approval. Revised portion are marked in red in the paper. The main corrections in the paper and the responds to yours comments are as following:
Question,
The explanations of the authors are consistent but are not reflected in their manuscript. None of the questions and suggestions from the first review were considered. The changes that were made are fundamentally format.
Response,
We are very sorry for our negligence of the revised manuscript. We have revised our manuscript carefully according to the suggestion from the first review. The reason maybe that the “track changes” could be not displayed in the PDF file after we upload the revised manuscript with modification trace using the “track changes” function using MS Word. Considering the reviewer’s suggestion, we have revised the title and related content in main text of this manuscript, and supplemented the correlation analysis and discussion of variation in the production of walnut oil components and climatic variables. The revised portion are marked in red in the revised manuscript. Please review line130-133 in the section of materials and methods, line236-244 and Figure 5 in the section of results, line283-286, line298-302, line314-316, and line332-336 in the section of discussion, and etc.
Special thanks to you for your good comments.
We tried our best to improve the manuscript and made some changes in the manuscript. We appreciate for yours warm word earnestly, and hope that the correction will meet with approval.
Once again, thank you very much for your comments and suggestions.

Reviewer 3 Report
Dear Authors,
Thank you very much for the accurate answers, explanation and proofreading of the text in accordance with the comments of the reviewer.
In my opinion still the problem lies in the two-way ANOVA. I would like to refer to your explanations “Response: Three species/genotypes were selected in our study. Six hybrids cultivated in six different sites. Three J. regia cultivated in three different sites. Two J. sigillata cultivated in two different sites. In our experimental design, two J. sigillata and two hybrids cultivated in the same two sites (Jiulong and Leibo). Therefore, the data of 2 species × 2 sites was used in two-way ANOVA (two factors: 2 species and 2 sites).”
Rev: This description/information appeared for the first time; you and me know it. Unfortunately, such information did not appear in either the materials and method section or the results section, therefore text is enigmatic. Such information should be given in the materials and method section and the results section (in table title also) , and the description of the results should be clear.
Essential comments:
Rev: Sections 3.1. Fatty acid composition and 3.2. Micronutrient levels should be improved.
Line 114-128: 2.3. Micronutrients determination …
Rev: The polyphenols and flavone are micronutrients? In my opinion there are secondary metabolites.
Line 154-156: The content of linoleic (F = 22.62, P = 0.000) and linolenic acid (F = 13.60, P = 0.001) in hybrids (J. sigillata × regia) were significantly higher than common walnut (J. regia).
Rev: Such description is unfortunate. F-value it is ANOVA result and shows differences between all species. Statistically significantly differences between hybrids (J. sigillata × regia) and common walnut (J. regia) were confirmed by LSD test.
Identical remark to the next sentence.
Line 161-162: Notably, there was a negative correlation between the content of oleic and linoleic acid (r = -0.979, P = 0.000) (Figure 2).
Rev: Just one sentence on correlation? The correlations should be described in more detail.
Line 163-172:
Rev: Source/attachment should be cited.
Line 155: (F = 22.62, P = 0.000)
Rev: P = 0.000 suggests that p is 0 but but it's not true, it's just the output of the statistic program; realistically maybe it is P = 0.0004. In my opinion, in this and other similar cases (Line 158, 161, 162 …….), you should replace by P < 0.001.
Line 159: walnut. oleic acid
Rev: ?
Line 159-161: oleic acid was the major MUFAs, and the content of oleic acid in common walnut was significantly higher than that in hybrids (F = 30.35, P = 0.000) and iron walnut (F = 5.40, P = 0.026).
Rev: Value F is result of the ANOVA, one result for one parameter. In the sentence above we have a situation where 2 ANOVA results are given for one parameter. Which value F is correct? It may be safer excluded the ANOVA results.
Line 168-172: The results of two-way ANOVA showed significantly differences of linolenic acid content between sites (F = 16.921, P = 0.003), and species (F = 8.001, P = 0.022). Significantly differences were also shown in fatty acids content between cultivars in different site, such as MUFA (F = 50.73, P = 0.000) and PUFA (W 171 = 15.316, P = 0.009).
Rev: The phrase above confirms that two-way ANOVA was done for all sites. Moreover, “species”?, “cultivars”?; There are species or cultivars or hybrids or species/genotypes….? . Consequently, the nomenclature must be maintained.
Line 175: … significant different at the level of 0.05.
Rev: Which one level? Should be corrected.
Line 189-190: The flavone content was ranged from 0.60-5.78 mg/kg, and showed similar trends with tocopherol.
Rev: Source/attachment should be cited. Informations from a wide variety of sources are provided.
Line 191-192: The content of flavone was not influenced by the genotypic variation or cultivation sites alone, but be affected by the interaction of species × site.
Rev: Source/attachment should be cited. Informations from a wide variety of sources are provided. What about polyphenols?
Line 191-192: Significantly differences (F = 192 2.775, P = 0.001) were obtained in the content of polyphenols quantified in the different 193 walnut species, and showed opposite trends with tocopherol and flavone. The polyphenols content of common walnut oil was significantly higher than that in hybrids walnut 195 oil (F = 17.63, P = 0.000), and 1.2 times higher than that in iron walnut oil (F = 1.187, P = 196 0.333).
Rev: Source/attachment should be cited. Informations from a wide variety of sources are provided.
Line 199-200: Figure 2. The correlation between chemical characterisctis of walnut oil Note: * , **, and *** indicated significant correlation at the level of 0.05, 0.01, and 0.001.
Rev: Which one level? Should be corrected.
Line 202: Figure 3. Micronutrient composition and contents of walnut oils
Rev: The polyphenols and flavone are micronutrients? In my opinion there are secondary metabolites.
Line 205: Table 3. Two-way analysis of variance (two-way ANOVA) for micronutrient
Rev: Title is very modest and enigmatic – information about statistic only. What is presented? All data or only 2 species x 2 sites? What species? What sites? These data should be added.
Line 230: above sea level (ASL)
Rev: Is above sea level a climatic factor?
Line 233: … correlation between latitude (LAT) …
Rev: Is latitude a climatic factor?
Line 250-254: Figure 5. Correlation analyses between climatic factors and chemical composition of walnut oil 250 Note: MinRH, minimum relative humidity (%), AvgRH, Average relative humidity (%), AvgTEM, 251 average temperature (°C), MaxTEM, maximum temperature (°C), MinTEM, minimum temperature 252 (°C), ASL, above sea level (m), PRE, precipitation (mm), LAT, latitude (°), SSD, sunshine duration 253 (h), the same as below.
Rev: Are above sea level and latitude a climatic factors? See Fig. 6 and e.t.c.
Author Response
List of Responses
Dear Reviewer:
Thanks again for your comments concerning our manuscript entitled “Chemical compositions in walnut (Juglans spp.) oil: combined effects of genetic and climatic factors” (Manuscript ID: forests-1745433). Those comments are all valuable and very helpful for revising and improving our paper, as well as the important guiding significance to our researches. We have studied comments carefully and have made correction which we hope meet with approval. Revised portion are marked in red in the paper. The main corrections in the paper and the responds to yours comments are as following:
Question1,
In my opinion still the problem lies in the two-way ANOVA. I would like to refer to your explanations “Response: Three species/genotypes were selected in our study. Six hybrids cultivated in six different sites. Three J. regia cultivated in three different sites. Two J. sigillata cultivated in two different sites. In our experimental design, two J. sigillata and two hybrids cultivated in the same two sites (Jiulong and Leibo). Therefore, the data of 2 species × 2 sites was used in two-way ANOVA (two factors: 2 species and 2 sites).”
Rev: This description/information appeared for the first time; you and me know it. Unfortunately, such information did not appear in either the materials and method section or the results section, therefore text is enigmatic. Such information should be given in the materials and method section and the results section (in table title also) , and the description of the results should be clear.
Response,
Special thanks to you for your good comments. It is really true as you suggested that the clear information about two-way ANOVA should be given in the materials and method section and the results section. As you suggested that, we have supplemented the information about two-way ANOVA in the statistical analysis of the materials and methods section, the results section, and the table title. For instance, “Further, two-way ANOVA based on the data of two sites test (Jiulong and Leibo) of two species (iron walnut and hybrids) was used to explore the effect of genotype, environment, and the genotype-environment interaction on the chemical composition and content of walnut oil (Table 1).” in the materials and methods section. Please review line130-133 in the section of materials and methods, line168-171, line194-196, line210-212 in the section of results.
Question 2,
Rev: Sections 3.1. Fatty acid composition and 3.2. Micronutrient levels should be improved.
Response,
We have revised the text of section 3.1 and 3.2 according to the reviewer’s suggestion and comments. Please review the sections 3.1. Fatty acid composition, and 3.2 Micronutrient and secondary metabolites levels in the results section.
Question 3,
Line 114-128: 2.3. Micronutrients determination …
Rev: The polyphenols and flavone are micronutrients? In my opinion there are secondary metabolites.
Response,
It is really true as you suggested that polyphenols and flavone are secondary metabolites. The polyphenols and flavone also had the function of nutritional and healthy in walnut oil. We are very sorry for our incorrect writing of the name of these two components, and we have revised the title of section 2.3 and other related sites. Please review line110, 200, 211, 228, and 230-231.
Question 4,
Line 154-156: The content of linoleic (F = 22.62, P = 0.000) and linolenic acid (F = 13.60, P = 0.001) in hybrids (J. sigillata × regia) were significantly higher than common walnut (J. regia).
Rev: Such description is unfortunate. F-value it is ANOVA result and shows differences between all species. Statistically significantly differences between hybrids (J. sigillata × regia) and common walnut (J. regia) were confirmed by LSD test.
Identical remark to the next sentence.
Response,
We are very sorry for our incorrect writhing. Considering the reviewer’s suggestion, we have deleted the F value in the text. Please review line 151-152.
Question 5,
Line 161-162: Notably, there was a negative correlation between the content of oleic and linoleic acid (r = -0.979, P = 0.000) (Figure 2).
Rev: Just one sentence on correlation? The correlations should be described in more detail.
Response,
Considering the reviewer’s suggestion, we have described the results of correlation analysis in more detail in the results section. Please review line158-162.
Question 6,
Line 163-172:
Rev: Source/attachment should be cited.
Response,
As reviewer suggested that we have supplemented the results of fatty acid of two-way ANOVA and cited in the text. Please review line162-172.
Question 7,
Line 155: (F = 22.62, P = 0.000)
Rev: P = 0.000 suggests that p is 0 but but it's not true, it's just the output of the statistic program; realistically maybe it is P = 0.0004. In my opinion, in this and other similar cases (Line 158, 161, 162 …….), you should replace by P < 0.001.
Response,
The P value was replaced by P Ë‚ 0.001 according to the reviewer’s suggestion. Please review line153, 156, 157, 173, and 221.
Question 7,
Line 159: walnut. oleic acid
Rev: ?
Response,
We are very sorry for our negligence of mistake, and we have revised this sentence as “…walnut. Oleic acid (C18:1) was…”. Please review line154.
Question 8,
Line 159-161: oleic acid was the major MUFAs, and the content of oleic acid in common walnut was significantly higher than that in hybrids (F = 30.35, P = 0.000) and iron walnut (F = 5.40, P = 0.026).
Rev: Value F is result of the ANOVA, one result for one parameter. In the sentence above we have a situation where 2 ANOVA results are given for one parameter. Which value F is correct? It may be safer excluded the ANOVA results.
Response,
We are very sorry for our incorrect writing. We have revised the F and P value according to the ANOVA results, as follow, “Oleic acid (C18:1) was the major MUFAs, and the content of oleic acid in common walnut was significantly higher than that in hybrids and iron walnut (F = 9.81, P Ë‚ 0.001).” Please review line154-156.
Question 9,
Line 168-172: The results of two-way ANOVA showed significantly differences of linolenic acid content between sites (F = 16.921, P = 0.003), and species (F = 8.001, P = 0.022). Significantly differences were also shown in fatty acids content between cultivars in different site, such as MUFA (F = 50.73, P = 0.000) and PUFA (W 171 = 15.316, P = 0.009).
Rev: The phrase above confirms that two-way ANOVA was done for all sites. Moreover, “species”?, “cultivars”?; There are species or cultivars or hybrids or species/genotypes….? . Consequently, the nomenclature must be maintained.
Response,
These results showed the differences of fatty acid among species. Considering the reviewer’s comments, we have re-written the results in the parts. Please review line149-151 and 171-175.
Question 10,
Line 175: … significant different at the level of 0.05.
Rev: Which one level? Should be corrected.
Response,
The different letters showed the results of multiple comparisons with the LSD test at P ≤ 0.05, and the different letters indicate significant differences (P Ë‚ 0.05) between different species/sites. We have revised it according to the reviewer’s comments. Please review line178, 195, 225, and 269.
Question 11,
Line 189-190: The flavone content was ranged from 0.60-5.78 mg/kg, and showed similar trends with tocopherol.
Rev: Source/attachment should be cited. Informations from a wide variety of sources are provided.
Response,
Considering the reviewer’s suggestion, we have cited the source in this sentence. Please review line216, 218, 222.
Question 12,
Line 191-192: The content of flavone was not influenced by the genotypic variation or cultivation sites alone, but be affected by the interaction of species × site.
Rev: Source/attachment should be cited. Informations from a wide variety of sources are provided. What about polyphenols?
Response,
Considering the reviewer’s suggestion, we have cited the source in this sentence. The content of polyphenols was significantly influenced by cultivation sites and the interaction of species × sites. We have supplemented the result about polyphenols in the results section. Please review line218-219.
Question 13,
Line 191-192: Significantly differences (F = 192 2.775, P = 0.001) were obtained in the content of polyphenols quantified in the different 193 walnut species, and showed opposite trends with tocopherol and flavone. The polyphenols content of common walnut oil was significantly higher than that in hybrids walnut 195 oil (F = 17.63, P = 0.000), and 1.2 times higher than that in iron walnut oil (F = 1.187, P = 196 0.333).
Rev: Source/attachment should be cited. Informations from a wide variety of sources are provided.
Response,
Considering the reviewer’s suggestion, we have cited the source in these sentences. Please review line216, 218, and 222.
Question 14,
Line 199-200: Figure 2. The correlation between chemical characterisctis of walnut oil Note: * , **, and *** indicated significant correlation at the level of 0.05, 0.01, and 0.001.
Rev: Which one level? Should be corrected.
Response,
“* , **, and *** indicated significant correlation (P Ë‚ 0.05, 0.01, and 0.001) between 19 chemical characteristics of walnut oil, respectively.” We have revised it in the text. Please review line225.
Question 15,
Line 202: Figure 3. Micronutrient composition and contents of walnut oils
Rev: The polyphenols and flavone are micronutrients? In my opinion there are secondary metabolites.
Response,
We are very sorry for our incorrect writing of the name of these two components, and we have revised it along the whole text. Please review line110, 200, 211, 228, and 230-231.
Question 16,
Line 205: Table 3. Two-way analysis of variance (two-way ANOVA) for micronutrient
Rev: Title is very modest and enigmatic – information about statistic only. What is presented? All data or only 2 species x 2 sites? What species? What sites? These data should be added.
Response,
The data of two cultivation sites (Jiulong and Leibo) of two species (iron walnut and hybrids) were used to two-way ANOVA. We are very sorry for our incorrect writing, and we have added the information of two-way ANOVA in the section of materials and methods, results, and tables. Please review line130-133 in the section of materials and methods, line168-171, line194-196, line210-212 in the section of results.
Question 17,
Line 230: above sea level (ASL)
Rev: Is above sea level a climatic factor?
Response,
We are very sorry for our negligence of the analysis of climatic factors. It is really true as reviewer comments that above sea level (ASL) is not a climatic factor. We have deleted the results related to ASL. Please review the section of 3.4. Climatic factors and fatty acid composition in results section, Figure 5, and Figure 6.
Question 18,
Line 233: … correlation between latitude (LAT) …
Rev: Is latitude a climatic factor?
Response,
We are very sorry for our negligence of the analysis of climatic factors. It is really true as reviewer comments that latitude is not a climatic factor. We have deleted the results related to latitude. Please review the section of 3.4. Climatic factors and fatty acid composition in results section, Figure 5, and Figure 6.
Question 19,
Line 250-254: Figure 5. Correlation analyses between climatic factors and chemical composition of walnut oil 250 Note: MinRH, minimum relative humidity (%), AvgRH, Average relative humidity (%), AvgTEM, 251 average temperature (°C), MaxTEM, maximum temperature (°C), MinTEM, minimum temperature 252 (°C), ASL, above sea level (m), PRE, precipitation (mm), LAT, latitude (°), SSD, sunshine duration 253 (h), the same as below.
Rev: Are above sea level and latitude a climatic factors? See Fig. 6 and e.t.c.
Response,
We are very sorry for our negligence of the analysis of climatic factors. It is really true as reviewer comments that above sea level (ASL) and latitude is not a climatic factor. We have deleted the results related to ASL and latitude. Please review the section of 3.4. Climatic factors and fatty acid composition in results section, Figure 5, and Figure 6.
Special thanks to you for your good comments.
We tried our best to improve the manuscript and made some changes in the manuscript. We appreciate for yours warm word earnestly, and hope that the correction will meet with approval.
Once again, thank you very much for your comments and suggestions.

Reviewer 4 Report
The authors should indicate how the two-way ANOVA was performed, based on their response to my comment.
Author Response
List of Responses
Dear Reviewer:
Thanks again for your comments concerning our manuscript entitled “Chemical compositions in walnut (Juglans spp.) oil: combined effects of genetic and climatic factors” (Manuscript ID: forests-1745433). Those comments are all valuable and very helpful for revising and improving our paper, as well as the important guiding significance to our researches. We have studied comments carefully and have made correction which we hope meet with approval. Revised portion are marked in red in the paper. The main corrections in the paper and the responds to yours comments are as following:
Question1,
The authors should indicate how the two-way ANOVA was performed, based on their response to my comment.
Response,
Special thanks to you for your good comments. It is really true as you suggested that the clear information about two-way ANOVA should be given in the materials and method section and the results section. As you suggested that, we have supplemented the information about two-way ANOVA in the statistical analysis of the materials and methods section, the results section, and the table title. For instance, “Further, two-way ANOVA based on the data of two sites test (Jiulong and Leibo) of two species (iron walnut and hybrids) was used to explore the effect of genotype, environment, and the genotype-environment interaction on the chemical composition and content of walnut oil (Table 1).” in the materials and methods section. Please review line130-133 in the section of materials and methods, line168-171, line194-196, line210-212 in the section of results.
Special thanks to you for your good comments.
We tried our best to improve the manuscript and made some changes in the manuscript. We appreciate for yours warm word earnestly, and hope that the correction will meet with approval.
Once again, thank you very much for your comments and suggestions.
